# Teamwork, Spirit of the Game and Communication: A Review of Implications from Sociological Constructs for Research and Practice in Ultimate Frisbee Games

**José Pedro Amoroso** [1,2,*], **Jay Coakley** [3], **Ricardo Rebelo-Gonçalves** [1,2,4], **Raul Antunes** [1,2,5],
**João Valente-dos-Santos** [6] and **Guilherme Eustáquio Furtado** [4,7,*]

1    Department of Human Kinetics, Polytechnic Institute of Leiria, 2411-901 Leiria, Portugal;
     ricardo.r.goncalves@ipleiria.pt (R.R.-G.); raul.antunes@ipleiria.pt (R.A.)
2    CIEQV, Life Quality Research Centre, Polytechnic Institute of Leiria, 2040-413 Leiria, Portugal
3    Department of Sociology, University of Colorado, Colorado Springs, CO 80918, USA; jcoakley@uccs.edu
4    CIDAF, The Research Unit for Sport and Physical Activity, Faculty of Sport Sciences and Physical Education,
     3040-248 Coimbra, Portugal
5    ciTechCare, Center for Innovative Care and Health Technology, Polytechnic of Leiria, 2411-901 Leiria, Portugal
6    CIDEFES, Centro de Investigação em Desporto, Educação Física e Exercício e Saúde, Universidade Lusófona,
     1749-024 Lisboa, Portugal; joao.valente.santos@ulusofona.pt
7    UICISA:E, Health Sciences Research Unit: Nursing, Nursing School of Coimbra, 3004-011 Coimbra, Portugal
*    Correspondence: jose.amoroso@ipleiria.pt (J.P.A.); guilhermefurtado@esenfc.pt (G.E.F.)

**Abstract:** Ultimate Frisbee (UF) is a non-contact, challenging, and self-promoted team sport. Its characteristics, such as the game environment and rules, appear to influence the on-the-pitch behaviour of players. This article examines the content of qualitative studies to determine if and how the unique characteristics of UF may be related to nine sociological themes, that may be identified during gameplay. These themes include the following: (a) competition and performance; (b) enjoyment; (c) communication; (d) cooperation/friendship; (e) behaviors/welfare; (f) teamwork/social skills; (g) environment/lifestyle; (h) rules/self-refereeing and (i) spirit of the Game (SOTG). The review was conducted according to PRISMA guidelines. A comprehensive search protocol was used to identify, screen, and select published research articles under a Qualitative Systematic Review (QSR). The search was occurred from 1 June to 30 December 2020 with no limitations regarding the year of publication. Original English-language papers that contained relevant data regarding sociological themes and UF were selected. As a result, nine papers were qualified to be included in the final version of QSR. The files analyzed were structured with MAXQDA. A total of 521 references were identified and selected for analysis. After the Screening (n = 301) and Eligibility (n = 71) phases, a total of 30 potential papers were selected and classified. Nine studies were included in the final analysis. The three most cited sociological themes in these studies were: communication, teamwork/social skills, and spirit of the game. Research suggests that UF involves patterns of interaction related to communication and the spirit of the game that encourage active lifestyles. Finally, we point out that UF is an appropriate sport to include in physical education classes in which the creation of positive relationships between students is the desired outcome. This topic should be explored further through interventional studies done in different contexts although the evidence suggests that UF offers players unique opportunities to experience a combination of physical activity and enjoyment.

**Keywords:** qualitative research; sport sociology; teamwork; spirit of the game; communication

## 1. Introduction

This paper focuses on knowledge production in sociology as it occurs through systematic research strategies designed to maximize objectivity (Bourdieu 1991). Our review of the research on Ultimate Frisbee (UF) indicates that the sport was first described by Clark, Hamilton, & Bowden in a 1981 article published in the *Journal of Physical Education*

*and Recreation.* In subsequent years, UF has been studied and described as an attractive alternative to traditional team sports in physical education classes. Additionally, a pedagogical sequence called UF Curriculum was developed for physical educators who wanted to introduce this sport modality to students in the United States (Caporali 1988). At the same time, experts have suggested that participation in UF helps to develop cognitive, psychomotor, and affective skills as well as general cardiovascular fitness (Caporali 1988; Clark et al. 1981).

Our review of the published literature also indicates that researchers in the sports sciences and related fields have become increasingly interested in studying the characteristics and impact of UF on participants. Studies have focused on: (i) physical, cardiovascular and metabolic demands in healthy adults and athletes (Krustrup and Mohr 2015; Leich et al. 2019; Weatherwax et al. 2015); (ii) gender differences among school and university players (Neville 2019; Piepiora et al. 2020); (iii) and throwing biomechanics, disc trajectory and injury prevention (Akinbola et al. 2015; Koeble and Seiberl 2020). Recent research in sport sociology has paid special attention to the normative foundations of UF as they are connected with the rules of the sport and the ethical orientations and competitiveness of participants (Crocket 2015, 2016; Griggs 2009b).

Research over the last four decades suggests that participation in UF provides valuable experiences that make it more than a mere alternative to traditional ball sports (Caporali 1988). Although more research is needed, existing studies report the following: *First*, the personality profile of the ultimate frisbee players is similar to the profile of players practicing other sports (Piepiora et al. 2020). *Second*, leadership behaviours among UF players foster acceptance of group goals and promote teamwork, players have high-performance expectations and focus on task cohesion; and the acceptance of group goals and the emphasis on teamwork is associated with social cohesion among players (Callow et al. 2009). *Third*, participation in Ultimate Frisbee has effects that carry over into everyday life in society (Guette et al. 2019). *Fourth*, the "spirit of the game" (SOTG), a cultural dimension of the sport, emphasizes self-enforcement of rules and respect for opponents and influences how players manage unethical actions and avoid a normative focus on winning at all costs (Griggs 2011). Our intent in this paper is to analyse selected qualitative studies to identify unique social dynamics associated with playing UF. However, the decisional complexity in motor games, confirms the differences among triads from the point of view of motor communication (Aguilar et al. 2018).

In this review of research on UF, it was decided that qualitative research would provide the most useful insights into the social dynamics of the game, the experiences of participants, and the usefulness of UF in physical education courses.

Qualitative research methods are widely used by sociologists. They involve collecting detailed information about specific people, groups, and situations; identifying patterns, unique features, and the meanings given to relationships and experiences; and analyzing information through the use of interpretive procedures and tests. Data are usually collected through systematic observations of particular social situations and events, and interviews designed to identify the meanings underlying the relationships and experiences of individuals participating in those situations and events. These data are systematically analyzed to provide detailed descriptions of what people feel, say, and do in the context of particular social situations and events.

Qualitative research methods are used when the goal is to discover the motives and meanings that underlie what people say and do, or when it is important to understand the precise details of what occurs in specific kinds of relationships, groups, and social contexts, such as playing a particular sport (Kuper et al. 2008). For example, qualitative methods might be used to discover and understand the conditions under which young people choose to play a sport, the meanings that they give to their sport experiences, and how those experiences are integrated into the rest of their lives. Sociologists frequently use qualitative methods when studying the social dynamics involved in sport participation, especially when participation occurs in a new or unique form of sport.

Collecting data through observations and interviews is time-intensive. The validity and reliability of data depend on the researcher being able to develop relationships so there is trust and rapport developed with the people being studied. The goal of sociologists who do observational research and conduct in-depth interviews is often to deepen or challenge existing knowledge about social phenomena or explore and present baseline information about social experiences, situations, and events about which little is known. This baseline information is then used to formulate subsequent research, both qualitative and quantitative, that studies particular social phenomena from multiple vantage points.

Because interpretation is a core feature of qualitative research, the researcher must be critically self-reflective during the entire research process. In practical terms, the researcher is a subject and an object in the research process. This does not destroy objectivity, but it challenges the researcher to be aware of their vantage point and relationship with what is being studied (Bourdieu 1991; Bourdieu and Wacquant 1992; Hill 2020; O'Brien et al. 2014).

We also chose to focus on qualitative research because we were concerned with two characteristics of UF that distinguish it from other sports: self-officiation and SOTG (Robbins 2004). Issued from the rules of the games, universals represent different social frameworks that call for individual motor action according to more or less permissive action logics (Parlebas 2020). To our knowledge, little research has focused on the sociological themes that may be characteristic when UF is played. Our objective in this qualitative systematic review (QSR) was to explore how the unique characteristics of UF are related to the following nine sociological themes as UF is played: (a) competition/performance; (b) enjoyment; (c) communication; (d) cooperation/friendship; (e) behaviors/welfare; (f) teamwork/social skills; (g) environment/lifestyle; (h) rules/self-refereeing and (i) SOTG.

## 2. Materials and Methods

The research team decided that to improve the accuracy of the search in the different databases, the search terms should be selected in advance. This is because the tools to assist meta-search change depending on each database. The keywords as agreed by the authors were the following: "frisbee", "flying disc", "frisbee" OR "flying disc", "frisbee" OR "Flying disc" AND "sports"; "ultimate frisbee"; "frisbee" OR "flying disc" OR "disco voador" and "ultimate frisbee" AND "sociology" (Table 1). The goal was to identify relevant articles in this conceptual realm. The inclusion criteria for these articles were studies published in English in peer-reviewed journals, and the search occurred from 1 June to 30 December 2020. Articles were searched across multiple academic disciplines (e.g., title, abstract, text) and each article was independently examined by (J.P.A.) and (G.E.F.) to assess its quality.

The quality of the articles was assessed by using evaluative criteria developed by members of the Evidence for Policy and Practice Information Co-ordinating Centre (EPPI-Center) and other specialists (Harden et al. 2004; Martins et al. 2014; Popay et al. 2006). Exclusion criteria were studies unrelated to the context of UF. Studies with no abstract available for screening and those not available in English translation were also excluded.

Finally, nine papers were qualified to be included in the final version of QSR. (Table 1) shows the key terms used in the respective databases during the first phase, considering the number of articles generated from the different entries with the isolated or combined terms. The Preferred Reporting Items for Systematic Reviews and Meta-Analyses (PRISMA) guidelines were applied while conducting the review (Liberati et al. 2009; Moher et al. 2015).

**Table 1.** Search terms used depending on the different databases and the number of articles generated in the pilot search.

| Data Bases | "Frisbee" | "Flying Disc" | "Frisbee" or "Flying Disc" | "Frisbee" or "Flying Disc" or "Disco Voador" | "Frisbee" or "Flying Disc" and "Sports" | "Ultimate Frisbee" | "Ultimate Frisbee" and "Sociology" |
|---|---|---|---|---|---|---|---|
| PUBMED (Mediline) | 260 | 6 | 262 | 1425 | 31 | 17 | 0 |
| WEB OF SCIENCE | 213 | 22 | 212 | 212 | 197 | 53 | 3 |
| APA Psycinfo | 43 | 2 | 45 | 68 | 43 | 15 | 1 |
| B-ON | 9821 | 235 | 10005 | 10041 | 9821 | 1963 | 494 |
| ERIC | 30 | 0 | 30 | 31 | 30 | 10 | 10 |
| SPORTDiscus | 92 | 6 | 96 | 102 | 94 | 48 | 6 |
| Psychology & Behavioral Sciences | 11 | 2 | 11 | 13 | 11 | 2 | 2 |
| Academic Search Complete | 335 | 11 | 342 | 880 | 338 | 42 | 5 |
| SCIELO | 5 | 5 | 5 | 11 | 3 | 0 | 0 |
| Cochrane database | 7 | 1 | 8 | 9 | 0 | 0 | 0 |
| Total of records | 10817 | 941 | 11016 | 12792 | 10568 | 2150 | 521 |

The analysis involved a process in which findings were identified, classified, and coded (O'Connor and Penney 2021). The first author (J.P.A.) conducted the coding analysis of the text. Analytic documentation refers to decisions made in coding, categorizing, and comparing data (Sandelowski and Barroso 2003). The second phase was to identify if any of the nine sociological themes in discussions of the social dynamics during the playing of Ultimate Frisbee. The second author (G.E.F.) made the verification process, coding a randomly selected subset of the selected papers, and undertaking additional checking of a sub-set of codes attributed within the other papers. The analysis was performed by (J.P.A.) using the software program MAXQDA® Analytics Pro 2020 software (release 20.03.0). The files analyzed were structured with MAXQDA (folder structure); text-specific overviews with appropriate coding, codes, and memos (Berlin 2008).

The first author (J.P.A.) was supervised by (G.E.F.) in the coding process. Regular twice-a-month meetings were conducted with the core study team (J.P.A., G.E.F.) over 6 months to reduce subjectivity. The analysis followed an inductive approach in which coding topics were derived directly from the text data. An interpretive approach sets the scene for the analysis; it shapes the choice of methodology, and it informs the questions which the researcher asks of the text (Bohnsack 2014). This process allowed us to uncover initial codes and to start conceptualizing thematic categories. According to the methodological characteristics listed in (Table 2), a fundamental premise of grounded theory is to allow the key issues to emerge rather than to force them into preconceived categories (Oktay 2012; Charmaz and Mitchell 1996).

In the third phase, codes were isolated by themes, within the initial codes such as attire, language, and relationship with the game. This was done to determine if playing UF had an impact on the following game characteristics (themes): (a) competition and performance; (b) enjoyment; (c) communication; (d) cooperation/friendship; (e) behaviors and welfare; (f) teamwork/social skills; (g) environment/lifestyle; (h) rules/self-refereeing and (i) spirit of the game.

**Table 2.** Methodological characteristics of the included studies in the review (n = 9).

| Reference Number/Author | Participants | Method | Type | Theme | Categories |
|---|---|---|---|---|---|
| 1. Robbins (2004) "That's cheap." The rational invocation of norms, practices, and an ethos in ultimate frisbee | 10 8 - | Informal Interviews Observation Media analysis | University competitive | Norms Ethos Social dilemmas in Ultimate Frisbee | (c) (d) (e) (f) (g) (h) (i) |
| 2. Griggs (2009b) 'When a ball dreams, it dreams it's a Frisbee': the emergence of aesthetic appreciation within Ultimate Frisbee | 20 | Interviews Observation Media analysis | Competitive Social participation Self-initiated | Examines aesthetic elements in Ultimate Frisbee | (b) (c) (d) (e) (f) (g) (h) (i) |
| 3. Griggs (2009a) 'Just a sport made up in a car park?': The 'soft' landscape of Ultimate Frisbee | 30 | Interviews Participant Observation Media analysis | Competitive Training Sessions Social Events | Examines Landscape Ethnographic approach | (a) (b) (c) (e) (f) (g) (h) (i) |
| 4. Griggs (2011) 'This must be the only sport in the world where most of the players don't know the rules': Operationalizing self-refereeing and the spirit of the game in UK Ultimate frisbee | 20 | Interviews and "list mining" Participant Observation Researching (internet forums) | Competitive Training Sessions Social Events | 'social contract' 'spirit of the game' viability of self-refereeing | (a) (b) (c) (e) (f) (g) (h) (i) |
| 5. Robbins (2012) Playing with fire, competing with spirit: Cooperation in the sport of ultimate | 1 team | interviews field-notes | Competitive (open division) | follow the norms and values unique to Ultimate. | (a) (b) (c) (d) (e) (f) (g) (h) (i) |
| 6. Crocket (2013) 'This is men's ultimate': (Re)creating multiple masculinities in elite open Ultimate Frisbee | 18 | interviews field-notes recording field notes Data sources | Elite (open division) training sessions National and International Tournaments | ethnographic approach | (a) (b) (c) (d) (e) (f) (g) (h) (i) |
| 7. Crocket (2015) Foucault, flying discs and calling fouls: Ascetic practices of the self in ultimate frisbee | 14 | semi-structured interviews textual analysis | Social and competitive tournaments | ethnographic research Foucauldian Ethics Ascetic practices | (a) (c) (d) (e) (f) (g) (h) (i) |
| 8. Crocket (2016) An ethic of indulgence? Alcohol, Ultimate Frisbee, and calculated hedonism | 14 | Interviews Media analysis | Competitive and social tournaments | Featherstone's concept ethnographic projectFoucault's ethics | (a) (c) (d) (e) (f) (g) (h) (i) |
| 9. Neville (2019) Dressed to play: An analysis of gender relations in college women's ultimate Frisbee | 27 | one-on-one interviews Transcription Coding | Practices Tournaments Social Events | ethnographic research Insight tensions within Ultimate Frisbee through exploring forms of dress | (a) (c) (d) (e) (f) (g) (h) (i) |

### 3. Results

A total of 521 references were identified through the database as illustrated in the flowchart presented in (Figure 1) in the first phase. Out of these, 163 references were excluded after reading the title and abstract, and replication. After applying these initial criteria, a total of 301 articles entered phase 2 of eligibility. Of these, 232 papers were later excluded. After the full text of articles was assessed, a total of 71 articles remained eligible, 41 of which were excluded, mainly because they used quantitative research methods. In the last phase of Inclusion, all authors decided that only articles that have sociological dimensions would be included in the final article, considering the previously presented concepts. As a result, the four-phase flow diagram identifies the final selection phases of the studies. Following the input of data into the software Mendeley version 1.19.8 all duplicates were deleted (n = 232) Two reviewers (J.P.A., G.E.F.) performed an analysis to assess the relevance of all articles. No disagreements occurred. Articles were eliminated based on inappropriate aims and domains.

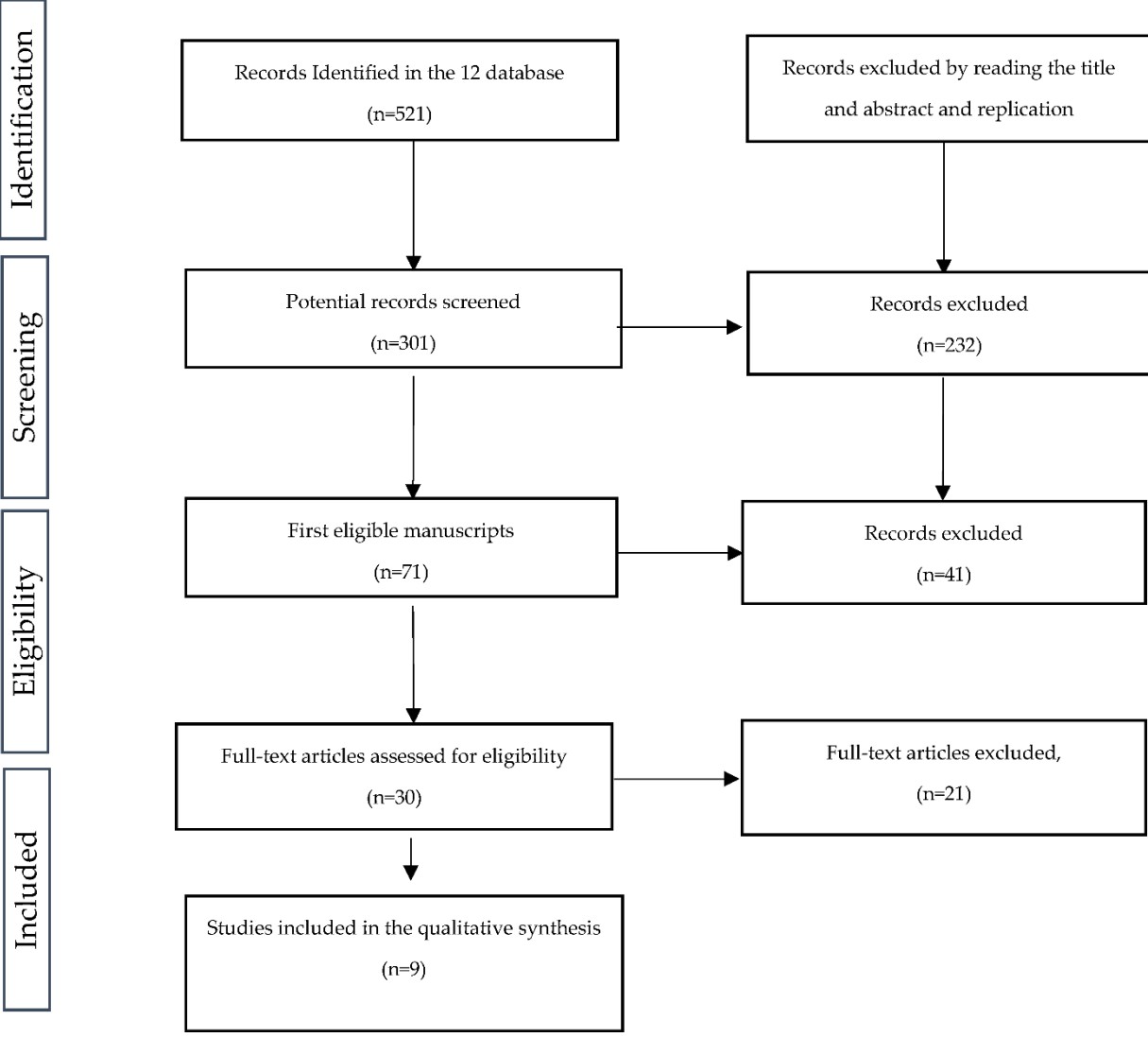

**Figure 1.** Flowchart of studies included following PRISMA guidelines.

In presenting findings, we note that much of the data relates to youth and adult participation in competitive and social tournaments. From the nine studies included in the analysis, we constructed a methodological interpretation of the studies included in the review in connection with the nine sociological themes (Table 3). After categorization of

the characteristics involved in playing UF, results emerge from categories present in the 9 studies: Behaviors and welfare, SOTG, Rules, and self-refereeing.

**Table 3.** Methodological interpretation of the included studies in the review (n = 9).

| Refs. Number/Author | (a) PC | (b) EN | (c) CO | (d) CF | (e) BW | (f) TSS | (g) EL | (h) RSR | (i) SE | Total |
|---|---|---|---|---|---|---|---|---|---|---|
| 1. Robbins (2004) | 0 | 0 | 40 | 7 | 19 | 36 | 14 | 25 | 30 | 171 |
| 2. Griggs (2009b) | 0 | 1 | 8 | 1 | 2 | 2 | 14 | 2 | 4 | 34 |
| 3. Griggs (2009a) | 17 | 17 | 8 | 0 | 2 | 8 | 10 | 1 | 2 | 65 |
| 4. Griggs (2011) | 11 | 2 | 8 | 0 | 7 | 12 | 2 | 18 | 19 | 79 |
| 5. Robbins (2012) | 14 | 4 | 4 | 23 | 8 | 7 | 1 | 12 | 18 | 91 |
| 6. Crocket (2013) | 16 | 2 | 6 | 6 | 7 | 11 | 8 | 4 | 4 | 64 |
| 7. Crocket (2015) | 5 | 0 | 13 | 7 | 27 | 12 | 5 | 20 | 20 | 109 |
| 8. Crocket (2016) | 1 | 0 | 3 | 1 | 3 | 13 | 11 | 3 | 2 | 37 |
| 9. Neville (2019) | 14 | 0 | 11 | 8 | 6 | 27 | 4 | 3 | 7 | 80 |
| | 78 | 26 | 101 | 53 | 81 | 128 | 69 | 88 | 106 | 730 |

Notes: PC = Performance competition; EJ = Enjoyment; CO = communications; CF = Cooperation and Friendship; BW = Behaviors and Welfare; TSS = Teamwork and Social Skills; EL = Environment and Lifestyle; RSR = Rules and Self- refereeing; SE = SOTG and Ethical.

The themes of Teamwork/Social Skills (126 references), SOTG (106 references), and Communication (101 references) emerged as the most frequently mentioned. UF is unique in that it involves self-refereeing, with players having to make and agree between themselves on decisions regarding rule infringements (e.g., contact fouls) (Callow et al. 2009). The emphasis on self-refereeing and SOTG in UF appears to be consistent with the social reasons why many young women choose to engage in sport (Spencer-Cavaliere et al. 2017).

The concept map showed that the Teamwork and Social Skills, SOTG and Ethical, Communication, Rules, and Self-refereeing, and Behaviors and Welfare were the most frequent social constructs presented in all analyzed papers. In addition, the intensity of the lines shows the potential relationship between the constructs, where the emphasis goes to the connections between Teamwork and Social Skills, SOTG, and Ethical and Communication (Figure 2).

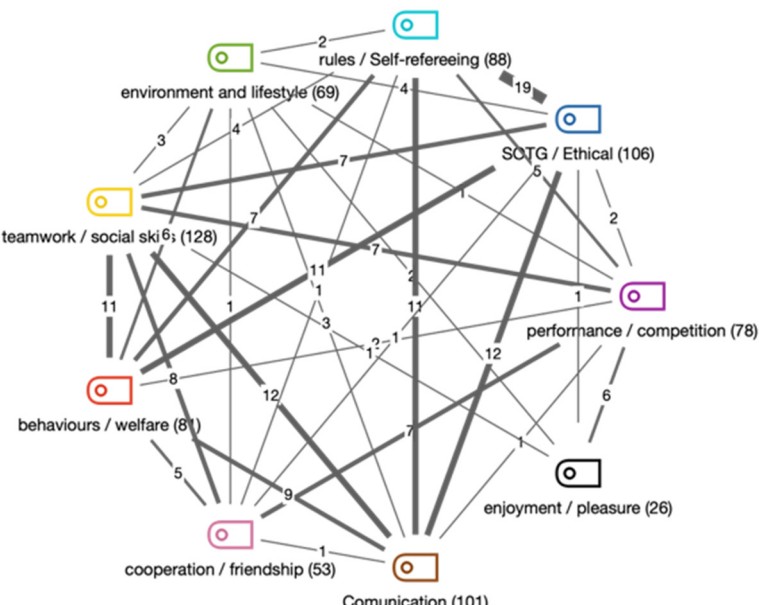

**Figure 2.** Conceptual map of interconnection between emerged categories.

Because UF is a self-refereeing sport, communication between players from different cultures is normal and depends on equivalence in thoughts and situations and not just

equivalence in expressions (Acquadro et al. 2008). Many regular social players were indifferent about developing their skills and showed little interest in playing the game at an advanced level (Griggs 2009b). Nevertheless, many regular social players evinced a high degree of commitment to the game and the people with whom they played regularly (Kerins et al. 2007). Crucial to the development of team performance were the mediating (teamwork) processes of communication and conflict management (Smith et al. 2013). All studies mentioned SOTG in different contexts but the ethos of SOTG is both a formalized and practiced part of UF (Spencer-Cavaliere et al. 2017).

All the categories were present in the majority of studies. Enjoyment was present in 5 studies (26 references) along with the assumption that rational thought interferes with feeling good (Bohnsack 2014). Competition/performance (78 references) was present in seven studies suggesting that even though UF is a team sport without referees, players are competitive, and team competitions differentially influence experiences related to performance, enjoyment, anxiety, and effort (Cooke et al. 2013). Behaviors/welfare (61 references) is also a theme present in all studies, even though others have noted that preoccupation with winning may be accompanied by a lack of concern for justice, fairness, and the welfare of others in competition (Lee et al. 2008).

Another important finding is that cooperation/friendship (53 references) was mentioned in seven studies indicating that because the sport is federally controlled and embedded in tiers of organizational constraint, competition is informally regulated through norms, reputations, and self-discipline respected by players (Robbins 2012). Also important in this respect is that leaders in the field of play can promote cooperation among followers and encourage them to work together towards a common goal (Callow et al. 2009). In the case of female UF participants, an emphasis on social interaction/friendship tends to disrupt the outcome and dominance-oriented structures embedded in traditional competitive sport (Spencer-Cavaliere et al. 2017).

Another influential theme, environment/lifestyle (69 references) has been emphasized by the World Flying Disc Federation and Ultimate Players Association (UPA) since the 1970s as a key feature of UF as it is played for enjoyment outside of the traditional school setting (Caporali 1988). In 2020 the #FRISBEELIFESTYLE emphasized that UF was organized around a lifestyle or an alternative way to play sport, but this has been given little attention in the research (Griggs 2011). This overlooks the fact that UF, like other fringe cultures, is linked with lifestyles and communities that are expressed through music, clothing, equipment, and locations that set them apart from the dominant culture (Gieseler 2019). This is important because experts often point out that a healthy lifestyle is an active process that first emerges during ontogenesis, especially during the maturity phase during which the background of an activity along with its norms and values constitute the environment in which people live (Pomohaci and Sopa 2018).

## 4. Discussion

The objective of this research project was to summarize the findings of qualitative studies that focus on UF and nine selected sociological themes. Nine qualitative studies dealing with UF were identified through a systematic process. Although research on UF is scarce, the findings of the selected studies indicate that it has distinguishing social features such as self-refereeing, collective arbitration, self-regulation, and independent communication that make it a relatively unique team sport. For example, it is self-referred even at the world championship level, and players are expected to abide by a formal code of fair play that is constituted by the *SOTG*, one of the unique features of UF (Crocket 2015).

Furthermore, UF is often played by small groups that are part of a larger and spatially dispersed self-governed sporting community (Robbins 2012). This community embraces a very specific language centered on the SOTG and emphasizing the importance of responsibility, respect, and honesty among players (Robbins 2012). When UF is played informally among small groups of friends interested in having an enjoyable experience,

the official rules may be replaced by informal norms that fit the situation so that play can be continuous and creative (Griggs 2011).

Studies have also consistently supported the idea that the themes of behaviors/welfare, goal orientations/motivation for approach, or avoidance influence performance norms and other behaviors (Meira and Fairbrother 2018). Although teamwork/social skills are the theme most often represented in the nine studies, the data also reveal that these other themes foster the improvement of teamwork within small heterogeneous teams a feature that would be important to Physical Educators who are unsure about how to organize students with different skill abilities (Carpenter 2010).

Additionally, cooperation/friendship is a theme that is a systemic and institutionalized aspect of SOTG. They are the foundation for an ethos comprised of an overarching system of norms that permeates all divisions and levels of competition. As a result, UF takes on the characteristics of a moral community organized around the spirit of the game (Robbins 2012). The systemic approach to rules combined with self-refereeing encourages young people to strive for personal excellence and competitive success at the same time that they value fairness and respect for both the rules and their opponents (Lee et al. 2008). Overall, the environment and culture of UF are closely linked with the sport's origins as part of 'the alternative sports movement' of the 1960s (Bale 1994; Griggs 2009a).

## 5. Conclusions

This review indicates that UF can help physical educators as they teach their classes and seek a strategy that promotes a commitment to communication and the norms linked to the SOTG. This makes it possible to facilitate the formation of a unique culture sustained by the student players. The nine studies that we have reviewed in this project provide strong support for creating positive social dynamics among players. We believe the findings of this study support social dynamics for UF participants. Alternatively, if we consider the socialization factors, that may lead to differing athletes, then we recognize that advantageous behaviors can be developed. Doing so is a much more meaningful and promising view of group dynamics in sport. Additional sociological research is needed to further investigate the ways that the nine social themes are a part of playing UF in different contexts. We presented directions for future research that appear new and particularly interesting.

**Author Contributions:** Conceptualization: J.P.A., R.R.-G., R.A., J.C., J.V.-d.-S., G.E.F.; Data curation: J.P.A., R.R.-G., J.V.-d.-S., G.E.F.; Formal analysis: J.P.A., R.R.-G., J.V.-d.-S., G.E.F.; Funding acquisition: J.P.A., J.V.-d.-S.; Investigation: J.P.A., R.R.-G., R.A., J.C., J.V.-d.-S., G.E.F.; Methodology: J.P.A., R.R.-G., R.A., J.V.-d.-S., G.E.F.; Project administration: J.P.A., J.V.-d.-S., G.E.F.; Resources: J.P.A., R.R.-G., R.A., J.C., J.V.-d.-S., G.E.F. All authors have read and agreed to the published version of the manuscript.

**Funding:** This work was supported by the Portuguese Foundation for Science and Technology, I.P., Grant/Award Number UIDB/04748/2020.

**Institutional Review Board Statement:** Not applicable.

**Informed Consent Statement:** Not applicable.

**Data Availability Statement:** The raw data supporting the conclusions of this article will be made available by the authors, without undue reservation.

**Acknowledgments:** The authors would like to thank the WFDF—World flying disc federation, to Polytechnic Institute of Leiria, and to CIEQV—Life Quality Research Center for all their support.

**Conflicts of Interest:** The authors declare no conflict of interest.

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
