# Peer review of "Teamwork, Spirit of the Game and Communication: A Review of Implications from Sociological Constructs for Research and Practice in Ultimate Frisbee Games"

_socsci, doi:10.3390/socsci10080300_

Round 1

Reviewer 1 Report

the authors of this manuscript qualitatively reviewed previously published articles on ultimate frisbee sport. This subject matter, frisbee or flying disc sport has been widely accepted as recreational sport in global sport industry but no critical knowledge on the positive effects or characteristic of UF have not been systematically carried by sport social scientist. This study adopted simple qualitative based analysis method following PRISMA guidelines but I can find must of merits from consuming this study. The followings are my judgement on this study: 

  • Simple but strictly followed PRISMA guideline. Therefore, you provide enough evidence on validity issues of qualitative based study
  • For table, you please work on title statement - make it more solid and precise still includes redundant information 
  • also for table 1, instead of presenting counting numbers of the keywords you should come up with better interpretation of those for your potential readers. perhaps proportion of those words might be more significant and meaningful considering Table 1, 2, and 3. 
  • Edit 1st row and column of Table 3. 
  • I believe the findings of this study support social dynamics for participants of this sport.  

Author Response

General Comments:

the authors of this manuscript qualitatively reviewed previously published articles on ultimate frisbee sport. This subject matter, frisbee or flying disc sport has been widely accepted as a recreational sport in the global sports industry but no critical knowledge on the positive effects or characteristic of UF have not been systematically carried by sport social scientist. This study adopted a simple qualitative-based analysis method following PRISMA guidelines but I can find must of merits from consuming this study. The followings are my judgment on this study: 

Dear Revisor,

A revised version of the manuscript now titled " TEAMWORK AND Communication: a review of implications from Sociological constructs for research and practice in Ultimate Frisbee games” was prepared and the authors are grateful for the comments.  Below it is possible to contact with a reply that was organized comment per comment. In addition, changes in the manuscript were highlighted in RED font.

Moderate English changes required

R: Possibly it will be associated with the American nationality of the co-author (Jay Coakley)

  • Simple but strictly followed PRISMA guidelines. Therefore, you provide enough evidence on validity issues of qualitative based study

R: Thanks.

  • For the table, you please work on the title statement - make it more solid and precise still includes redundant information 

R: Done, thanks.

  • also for table 1, instead of presenting counting numbers of the keywords, you should come up with a better interpretation of those for your potential readers. perhaps a proportion of those words might be more significant and meaningful considering Table 1, 2, and 3. 

R: We understand and appreciate the collaboration, however, our intention was to demonstrate rigor in the use of search terms in different databases. Thanks.

  • Edit 1st row and column of Table 3. 

R: Done, thanks.

  • I believe the findings of this study support social dynamics for participants of this sport.  

R: This comment was appreciated.  The revised version of the manuscript already incorporated the topic in the conclusion section.

We thank the reviewer for the insightful comments. We welcome the opportunity to clarify these important components of the manuscript. Comments and suggestions were essentials contributors to improve the quality of this manuscript.

Reviewer 2 Report

  1. Introduction

In introduction should be defined to notions of 'sport' and 'games', 'traditional sporting games', 'motor games' ..... These references should be very useful. Both are related to the motor comunication.

Pic, M., Navarro-Adelantado, V., and Jonsson, G. K. (2018). Detection of ludic patterns in two triadic motor games and differences in decision complexity. Front. Psychol. 8:2259. doi: 10.3389/fpsyg.2017.02259

Parlebas P (2020) The Universals of Games and Sports. Front. Psychol. 11:593877. doi: 10.3389/fpsyg.2020.593877

  1. Materials and Methods

It would be very interesting if you could include the correlation coefficients between the variables in table 3

  1. Discussion

The discussion should be more elaborate with the new approach to correlation.

Author Response

Comments and Suggestions for Authors

  1. Introduction

In introduction should be defined to notions of 'sport' and 'games', 'traditional sporting games', 'motor games' ..... These references should be very useful. Both are related to the motor comunication.

Pic, M., Navarro-Adelantado, V., and Jonsson, G. K. (2018). Detection of ludic patterns in two triadic motor games and differences in decision complexity. Front. Psychol. 8:2259. doi: 10.3389/fpsyg.2017.02259

Parlebas P (2020) The Universals of Games and Sports. Front. Psychol. 11:593877. doi: 10.3389/fpsyg.2020.593877

R: References were added, and it is believed that the introduction was improved

  1. Materials and Methods

It would be very interesting if you could include the correlation coefficients between the variables in table 3

R: This comment was appreciated. Not using the correlation denotes a classical conservatism of social knowledge. Our intention is not to make the study too statistical, however the program generates a mathematical output. The generated matrix allows us to assume in the qualitative field, with some care, but it does not allow us to analyze the results as quantitative outputs.

  1. Discussion

The discussion should be more elaborate with the new approach to correlation.

R: This comment was appreciated.  The revised version of the manuscript an infographic has been added to help the reader better understand the tables and discussion of the article.

We thank the reviewer for the insightful comments. We welcome the opportunity to clarify these important components of the manuscript. Comments and suggestions were essentials contributors to improve the quality of this manuscript.
